# *Annona cherimola* Seed Extracts Trigger an Early Apoptosis Response and Selective Anticlonogenic Activity against the Human Gastric Carcinoma Cell Line SNU-1

**DOI:** 10.3390/molecules28196906

**Published:** 2023-10-02

**Authors:** Johan Macuer-Guzmán, Claudia Giovagnoli-Vicuña, Giuliano Bernal, Lorena Lobos-González, Erwin de la Fuente-Ortega, Michael Araya-Castillo, Cristian Ibáñez

**Affiliations:** 1Laboratorio de Silvigenómica y Biotecnología, Departamento de Biología, Facultad de Ciencias, Universidad de La Serena, Avenida Raúl Bitrán 1305, Casilla 599, La Serena 1700000, Chile; johan.macuer@udalba.cl; 2Facultad de Ciencias Agropecuarias, Universidad del Alba, Cuatro Esquinas 060, La Serena 1700000, Chile; 3Departamento de Química Inorgánica, Facultad de Química y de Farmacia, Pontificia Universidad Católica de Chile, Avenida Vicuña Mackenna 4860, Macul, Santiago 7810000, Chile; ccgiovagnoli@uc.cl; 4Laboratorio de Biología Molecular y Celular del Cáncer (CáncerLab), Departamento de Ciencias Biomédicas, Facultad de Medicina, Universidad Católica del Norte, Larrondo 1281, Coquimbo 1781421, Chile; gbernal@ucn.cl; 5Centro de Medicina Regenerativa, Facultad de Medicina-Clínica Alemana, Universidad del Desarrollo, Avenida las Condes 12438, Lo Barnechea, Santiago 7710162, Chile; llobos@udd.cl; 6Laboratorio de Estrés Celular y Enfermedades Crónicas no Transmisibles, Núcleo de Investigación en Prevención y Tratamiento de Enfermedades Crónicas no Transmisibles (NiPTEC), Departamento de Ciencias Biomédicas, Facultad de Medicina, Universidad Católica del Norte, Coquimbo 1781421, Chile; edelafuente@ucn.cl; 7Centro de Investigación y Desarrollo Tecnológico de Algas (CIDTA), Facultad de Ciencias del Mar, Universidad Católica del Norte, Larrondo 1281, Coquimbo 1781421, Chile; mmaraya@ucn.cl; 8Instituto Multidisciplinario de Investigación y Postgrado, Universidad de La Serena, Avenida Raúl Bitrán 1305, Casilla 599, La Serena 1700000, Chile

**Keywords:** fruit waste, cancerous cells, ethanolic extract, bioactivity

## Abstract

The aim of this study was to evaluate, for the first time, the antiproliferative, apoptotic and diminishing effects of the anchored growth-independent capacity of an ethanol macerate extract from the *Annona cherimola* seed (EMCHS) in the human gastric cancer cell line SNU-1. The cells treated with EMCHS (20 μg/mL) significantly reduced the capacity to form clones of the tumor cell. Moreover, 50 μg/mL of EMCHS extract induced apoptosis, as was shown by the Annexin-V assay. UHPLC-MS/MS analysis detected two acetogenins (*Annonacinone* and *Annonacin*) in the EMCHS, which could be largely responsible for its selective antiproliferative effect. The identification of fatty acids by GC-FID showed the presence of eight fatty acids, among which was, oleic acid, which has recognized activity as an adjuvant in antitumor treatments. Taken together, our results indicate that the EMCHS seems promising for use as a natural therapy against gastric cancer disease.

## 1. Introduction

Gastric cancer (GC) ranks as the fifth most frequently diagnosed cancer and the third leading cause of cancer-related mortality worldwide [1]. Common risk factors for gastric cancer include Helicobacter pylori infection (HP), smoking, high-salt diets, and susceptibility to hereditary gastric cancer syndrome [2].

Currently, the main methods of cancer treatment are surgery, chemotherapy, or radiotherapy, either individually or in combination [3]. Chemotherapy typically involves metabolites that interact with DNA or affect proteins involved in DNA replication and cell cycle control. Moreover, monoclonal antibody-based therapies have been developed, targeting specific receptors in tumor cells, such as EGFR, VEGF, or PD-L1, to impede tumor growth [4]. Nevertheless, a significant challenge with these drugs is their non-selectivity, as they act on both cancerous and healthy cells, leading to side effects like hair loss, drug resistance, gastrointestinal lesions, neurotoxicity, nephrotoxicity, or bone marrow suppression [5].

In pursuit of alternative treatments with fewer or undetectable side effects on normal cells, extracts and compounds from plants, including the Annonaceae family, have shown selective cytotoxic effects against cancer cells [6]. Particularly, acetogenin metabolites found in the Annona species act at the mitochondrial level by interfering with electron transfer, inhibiting cellular respiration and inducing apoptosis [6]. For instance, the acetogenin annomocherin (present in *Annona cherimola*) exhibits potent selective cytotoxicity against breast carcinoma cell lines (MCF-7) and kidney carcinoma (A-498) [7]. Previously, our group discovered that an ethanolic extract obtained from the seeds of *A. cherimola* demonstrated selective antitumor effects on gastric cancer cells AGS [8]. Therefore, the objective of this study was to assess, for the first time, the apoptotic and clonogenic potential of an extract from *A. cherimola* in the human gastric cancer cell line SNU-1.

## 2. Results and Discussions

### 2.1. EMCHS Induces Apoptotic Cell Death in SNU-1

In order to identify if the EMCHS extract is capable of inducing cell death by apoptosis, we used an immunofluorescence assay to discriminate between apoptosis (Annexin-V) or necrosis (propidium iodide) cell death. As shown in Figure 1, 50 μg/mL of EMCHS extract induced a significant increase in the Annexin-V label of SNU-1 cells (green label, Figure 1(Aa)) compared with the controls (Figure 1(Ae)) and cells without labeling with propidium iodide (Figure 1(Ab)). This result shows that apoptosis, but not necrosis, is involved in the cell death triggered by EMCHS extracts. These results agree with other studies showing that extracts obtained from the genus *Annona* present apoptotic activity in tumor cells [3].

### 2.2. EMCHS Extract Decreased Anchored Growth Independent Capacity of the Tumorigenic Gastric Cells SNU-1

A colony formation assay, or clonogenic assay, is an in vitro cell survival assay used to examine the capability of a single cell to grow into a large colony through clonal expansion. We used the clonogenic activity to determine the effectiveness of the EMCHS extract as a cytotoxic agent. We observed a significant reduction in the formation of clones of SNU-1 cells as the concentration of EMCHS increased (compared to untreated cells, NT). At a concentration of extracts of 20 μg/mL, SNU-1 cells were practically not forming colonies. On the contrary, at the same concentration, more than double the colonies of normal gastric epithelial cells GES-1 were observed, evidencing an anticlonogenic selectivity of cancer cells by the EMCHS extract (Figure 2). This assay is consistent with that previously reported regarding the selectivity of the extracts and/or compounds from the genus *Annona* [9]. Furthermore, this inhibition of colony formation has been previously observed in non-melanoma skin cancer cell lines exposed to the Graviola (*Annona muricata*) extract [10]. In summary, the EMCHS extract decreased the clonogenic capacity of the SNU-1 cells and, at the same concentration, kept a greater number of surviving cells (with conserved reproductive integrity) of the normal gastric cell line GES-1.

### 2.3. The EMCHS Extract Contains Acetogenins and Several Types of Fatty Acids

Using the UHPLC-MS/MS approach, a chromatography separation (UHPLC) of the EMCHS extract was performed in full scan mode (Appendix A), followed by a search for the different masses of acetogenins. From this analysis, the possible presence of two acetogenins, annonacin (Pubchem, exact mass: 596.4652 g/mol) and annonacinone (Pubchem, exact mass: 594.4455 g/mol) was detected. Then, an MS/MS fragmentation of the respective experimental masses of molecular ion sodium adducts 617.4380 and 619.4538 was performed, and these results and their mass spectra are shown in Table 1 and Appendix A. Both the MS/MS spectra coincide with those found by other authors [10] for annonacin and annonacinone. Both acetogenins have a similar fragmentation pattern, with the same specific loss of γ-methyl-γ-lactone (−112 amu) (Table 1) [11]. As mentioned above, acetogenins have antiproliferative properties in cancer cells, inhibiting the mitochondrial complex I [10,12]. Thus, it is likely that these compounds play an important role in the antiproliferative activities of the *A. cherimola* EMCHS extract. In addition, several fatty acids have been identified to have antiproliferative activity. For example, oleic acid possesses antiproliferative activity in cancer cells [13] and has shown synergy with trastuzumab (Herceptin™), improving the inhibitory activity over breast cancer cells [14]. In our study, using gas chromatography, we identified eight fatty acids, including oleic acid (Table 2 and Figure 3), suggesting that in the antiproliferative activity of EMCHS extract, this fatty acid might be acting as an adjuvant.

## 3. Materials and Methods

### 3.1. Sample Preparation

*Annona cherimola* Mill. (Concha Lisa variety) was collected from the Chirimoyo garden belonging to the University of La Serena, Chile (29°54′44.9″ S; 71°14′41.9″ W). Five days after harvesting, seeds were collected from fully ripe fruits, washed, and dried in a heating oven (Binder, Tuttlingen, Germany) using natural convection at 35 °C for 2 days. The seeds were stored at −80 °C until use.

### 3.2. Ethanol Extract by Maceration from A. cherimola Seeds

To obtain the extract, 20 g of dry seeds were ground and macerated in absolute ethanol (100 mL) for 4 days in the dark. The macerate extract was filtered with Munktell filter paper (diameter 119 mm; grade 292) and a Sartorius filter (0.45 μm pore size). The extract was dried in a rotary evaporator (Büchi RE12, Diemtigen, Switzerland) under reduced pressure at 35 °C and centrifuged at 2000 RPM for 2 min. The precipitate (EMCHS) was recovered and dissolved in aqueous ethanol (1%) at 2 mg/mL. The EMCHS extract was stored at −80 °C.

### 3.3. Cell Culture

The SNU-1 cells (human gastric carcinoma cell line obtained from the laboratory of Dr. Bernal, CáncerLab, Coquimbo, Chile) were maintained in RPMI medium (Corning, Corning, NY, USA), while the GES-1 cells (human gastric epithelial cell line used as control cells, kindly donated by Dr. Dawit Kidane-Mulat, University of Texas at Austin, and stored in CáncerLab), were maintained in DMEM medium (Corning, Corning, NY, USA). Both culture media were supplemented with 10% fetal bovine serum (Hyclone, Logan, UT, USA) and 1% antibiotics (100 U/mL of penicillin and 100 μg/mL of streptomycin) (Corning, Corning, NY, USA). Cells were incubated in a CO_2_ incubator (Shel Lab, Cornelius, OR, USA) under 37 °C with 5% CO_2_ and 95% humidity.

### 3.4. Apoptosis Assay: Annexin-V/Hoechst/PI

To detect the presence of apoptosis in the SNU-1 cell, induced by the EMCHS extract, we used the Annexin-V AlexaFluor 488 assay (Invitrogen, Carlsbad, CA, USA) [15]. The SNU-1 cells (3 × 10^5^ cells/well) were cultured in 24-well plates (Costar, Washington, DC, USA) and grown overnight with 500 µL of RPMI 1649 medium (Sigma, Ronkonkoma, NY, USA) with 10% fetal bovine serum (GIBCO, Billings, MT, USA) and 1% antibiotics, as was mentioned. After 24 h, the cells were incubated with 50 µg/mL EMCHS for 3 h. Cisplatin at 7.5 μg/mL (Abcam, Cambridge, MA, USA) was used as a positive control at the same conditions as the treatment. After this time, the cells were transferred to Eppendorf tubes and centrifuged at 600× *g* for 3 min, and the supernatant was discarded. Cells were then washed twice with cold Annexin binding buffer (ABB) (10 mM HEPES, 140 mM NaCl, and 2.5 mM CaCl_2_, pH 7.4), and each time centrifuged for 3 min at 600× *g*. The final precipitate was dissolved and blocked with cold BSA (1%) in ABB for 30 min at 4 °C. Then it was centrifuged, and the supernatant was discarded. The cells were incubated with 80 μL of the fluorophore mixture: Annexin-V (1:5000) (Invitrogen, Carlsbad, CA, USA), propidium iodide (1 μg/mL) (Invitrogen), and Hoechst 33342 (1:2000) (Invitrogen, Carlsbad, CA, USA) for 60 min at 4 °C. Then, it was centrifuged at 600× *g* for 6 min and the supernatant was discarded. The cells were washed with 200 mL of ABB and precipitated under the same conditions as above. Subsequently, the cells were fixed with 200 μL for 15 min with 4% PFA (paraformaldehyde) at room temperature, then centrifuged at 600× *g* for 6 min. The supernatant was discarded, and 200 μL of ABB was added to the precipitate. The cells were centrifuged again at 600× *g* for 6 min, discarding the supernatant. The precipitate was brought to a final volume of 100 μL of ABB, and from this solution, 1 drop was added to a glass slide, dried for 5 min, and 10 μL of ProLong (Invitrogen, Carlsbad, CA, USA) was added. Subsequently, a coverslip was placed on the glass slide sample. The images of the samples were processed with Zeiss Laser Scanning Confocal Microscope (LSCM-800) (Carl Zeiss, Oberkochen, Germany) using 405, 488, and 561 nm lasers and the Plan-Apochromat 63×/1.46 oil immersion objective. The confocal image stack was acquired at 512 × 512 and processed to produce regions of interest (ROI) with approximately 10 cells/ROI. Finally, we quantified the apoptotic cells expressed as a ratio (numbers of positive AnnexinV cells/number of nucleus cells), the number of cells was determined using the software ZEN-2.1 modulo image analysis. The number of positive Annexin V cells was identified in channel 488 nm (over the threshold of 11,500), and the number of nuclei in channel 405 nm.

### 3.5. Anchored Growth Independent Assay

A 12-well culture plate was pre-prepared with 1 mL of 0.5% agarose in RPMI 10% FBS. The SNU-1 and GES-1 cells were seeded (2 × 10^3^ live cell/well, Appendix A) on the previously prepared agarose bed and grown at 37 °C with 5% CO_2_. At 24 h after this seeding, each well received 200 μL of RPMI medium without serum and under several concentrations of EMCHS to be evaluated. The addition of medium (without extract) was repeated on days 4, 8, and 10, to avoid agar dehydration. The culture was monitored every 2 days and colony formation was observed. The formation of colonies was scored under a phase-contrast microscope at 10×. At day 14, the colonies with a diameter of approximately 100 μm were considered as viable clones.

### 3.6. Analysis of EMCHS Extract by UHPLC-MS/MS

The instrumental analysis used to identify organic compounds in the EMCHS extract was developed using a Dionex Ultimate 3000 UHPLC system (Thermo Fisher Scientific, Sunnyvale, CA, USA) coupled to an HRMS mass spectrometer (Q Exactive focus with an Orbitrap^®^ detector, Thermo Scientific). A Hypersil GOLD TM column 100 × 2.1 mm, 3 µm was used as stationary phase. The sample was eluted using a mobile phase formed by solvent A (H_2_O acidified with 0.1% formic acid) and solvent B (acetonitrile acidified with 0.1% formic acid), with a flow rate of 0.1 mL/min. The oven temperature was 40 °C, and 10 µL of the sample was injected. The detection was carried out in electrospray positive ion mode [(+) ESI], recorded in full scan mode. Scan range: (250–700 *m*/*z*); microscan: 1 scan/sec. The ESI conditions were as follows: sheath gas flow rate: 30, aux gas flow rate: 10, sweep gas flow rate: 0, spray voltage: 3.0 kV, capillary temperature: 325 °C, S-lens RF level: 100, aux gas heater temperature: 250 °C. Samples were measured in full scan mode, and later, the data were acquired in Selected Ion Monitoring (SIM) and data-dependent (ddMS2) acquisition mode. Confirmation for the *m*/*z* 617.4380 and *m*/*z* 619.4538 was carried out. The performance of fragmentation was at CE 35 eV and 50 eV, respectively.

### 3.7. Identification of Fatty Acids by GC-FID

The fatty acid profile was obtained by GC-FID. The extraction of lipids from extracts was performed using Folch reagent. The methyl esters of fatty acids (FAME) were obtained by methylation with 14% BF_3_ in MeOH (Merck, Darmstadt, Germany) for 10 min at 60 °C. The derivatized FAME were extracted by liquid–liquid extraction with hexane and washed with 20% NaCl. The organic phase was evaporated and resuspended in 1 mL hexane. The extract was filtered by PVDF 0.45 μm prior to injection. The FAME were analyzed by a Clarus 600 chromatograph, Perkin Elmer (Shelton, CT, USA), with an OmegaWaxTM320 capillary column (30 m × 0.32 mm × 0.25 μm). The carrier gas was helium at 1 mL/min, and the detection was made with FID (flame ionization detector). The temperature ramp started at 140 °C maintained for 5 min, and then increased to 240 °C at a rate of 2 °C/min. The injector was at 260 °C, and the volume injected was 1 μL. Supelco 37 Component FAME mix was used as a standard (Sigma-Aldrich, St. Louis, MO, USA). Chromatograms were analyzed using TCNAV software version 6.3.1.

### 3.8. Statistical Analysis

The data are presented as the mean ± SD of at least three independent experiments. The normality of the data was evaluated using the Shapiro–Wilk test, followed by a Bartlett test to evaluate the homogeneity of variance. To evaluate the significance of the differences between the averages of the treatments and the control, an analysis of variance (ANOVA) was used, followed by the post hoc Tukey multiple comparisons test. The significance was evaluated at three levels: * *p* < 0.05, ** *p* < 0.01 and *** *p* < 0.001.

## 4. Conclusions

This study demonstrates that the EMCHS extract of *A. cherimola* seeds is capable of triggering apoptosis at an early stage in gastric cells SNU-1. Additionally, the extract shows a decrease in the formation of clones of SNU-1 cells with increasing concentration, but this effect is not observed in normal gastric cells (GES-1). This specificity is essential for an extract with anticancer potential. The ethanolic extract was found to contain two acetogenins, annonacinone and annoancin, as well as oleic acid. We think that these compounds might be responsible for the demonstrated anticancer potential of the ethanolic extract. However, further in vitro and in vivo studies are needed to confirm that the EMCHS extract has the potential to be an effective anticancer agent. These results highlight the importance of evaluating the seeds of this species, which are currently considered a waste as a potential source of a natural pharmacological product with anticancer properties.

## Figures and Tables

**Figure 1 molecules-28-06906-f001:**
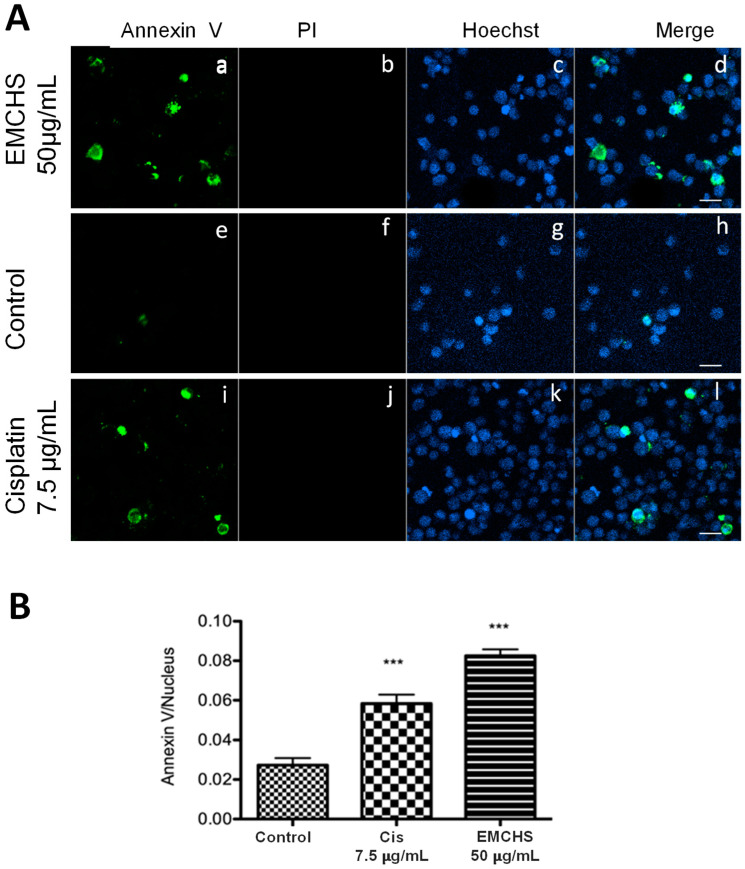
Ethanolic extract of EMCHS induces apoptosis in SNU-1. (**A**) The SNU-1 cells were cultured in RPMI medium for 3 h with the following: EMCHS 50 µg/mL (**Aa**–**Ad**), control (without treatment) (**Ae**–**Ah**), and Cisplatin 7.5 μg/mL (**Ai**–**Al**). The cells were labeled with Annexin-V (green), Hoechst 33342 (blue), and propidium iodide (red), and analyzed by confocal microscopy. The green color indicates apoptotic cells stained with Annexin-V. Scale bar 20 μm. (**B**) Quantification of apoptotic cells was according to the treatment performed. The asterisks represent the level of statistical significance, where *p* < 0.001 (***).

**Figure 2 molecules-28-06906-f002:**
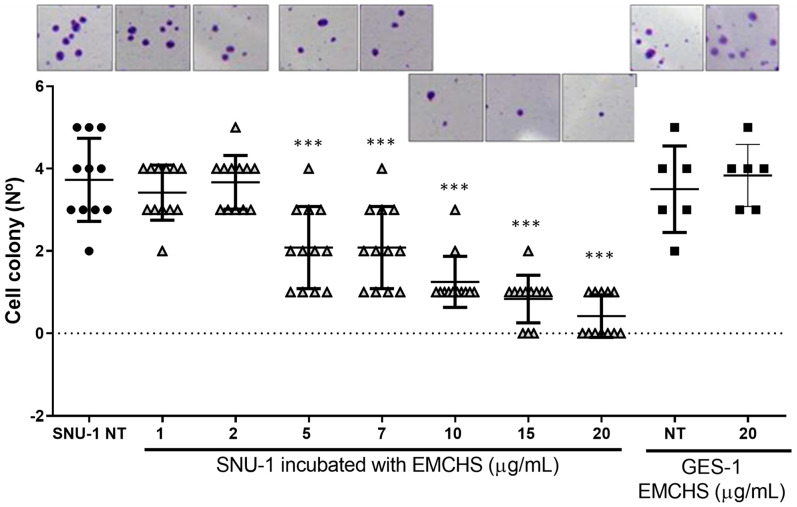
Anchored growth independent assay of SNU-1 cells exposed to EMCHS extract. After 14 days, colonies were counted (only clones over 100 nm were considered). The SNU-1 cells were treated with different concentrations of EMCHS extract. To evaluate the statistical significance of these results, the SNU-1 cells treated with EMCHS were compared to the control (no treatment, NT). The human gastric epithelial cell line (GES-1) was used as control cells. The asterisks represent the level of statistical significance, where *p* < 0.001 (***). Black circle SNU-1 no treated, black square GES-1 cells treated or not with EMCHS, grays triangle SNU-1 cells treated with different concentration of EMCHS.

**Figure 3 molecules-28-06906-f003:**
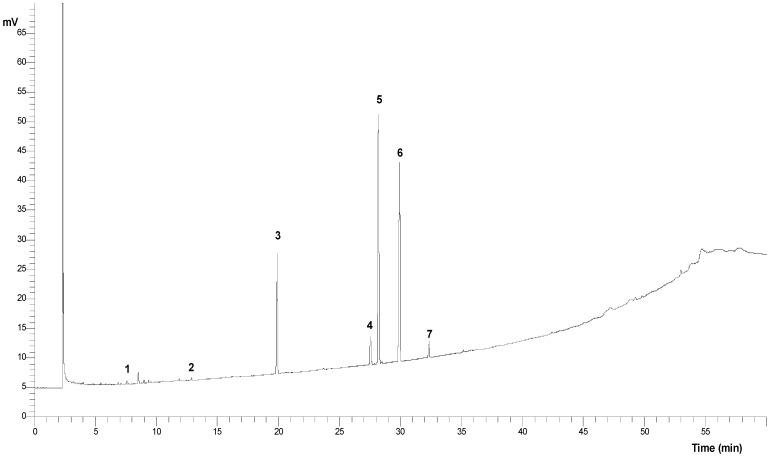
Chromatogram of fatty acids identified in ethanolic extract EMCHS. 1—Lauric acid, 2—Myristic acid, 3—Palmitic acid, 4—Stearic acid, 5—Elaidic acid, 5—Oleic acid, 6—Linoleic acid, and 7—Linolenic acid.

**Table 1 molecules-28-06906-t001:** Possible presence of acetogenins, analyzed by UHPLC-MS/MS.

Name	Retention Time (min)	Molecular Formula	*m*/*z* Experimental [M + Na]^+^	*m*/*z* Theoretical [M + Na]^+^	Error (ppm)	Fragment Ions
Annonacinone	6.55	C_35_H_62_O_7_	617.4380	617.4387	−1.13	505.3857 [M + Na − 112]^+^
Annonacin	6.73	C_35_H_64_O_7_	619.4538	619.4544	−0.96	507.4019 [M + Na − 112]^+^

**Table 2 molecules-28-06906-t002:** Fatty acids present in the EMCHS extract.

Fatty Acids Found in EMCHS Extract	Peak Number (Chromatogram Figure 3)	Molecular Formula	Reported Anticancer Activity	References
Lauric acid	1	C_12_H_24_O_2_	not	-
Myristic acid	2	C_14_H_28_O_2_	not	-
Palmitic acid	3	C_16_H_32_O_2_	not	-
Stearic acid	4	C_18_H_36_O_2_	not	-
Elaidic acid	5	C_18_H_34_O_2_	not	-
Oleic acid	5	C_18_H_34_O_2_	yes, GI50 = 10 µM	[13]
Linoleic acid	6	C_18_H_32_O_2_	not	-
Linolenic acid	7	C_18_H_30_O_2_	not	-

## Data Availability

Not applicable.

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
