# Peer review of "Annona cherimola* Seed Extracts Trigger an Early Apoptosis Response and Selective Anticlonogenic Activity against the Human Gastric Carcinoma Cell Line SNU-1"

_molecules, 2023, doi:10.3390/molecules28196906_

Round 1

Reviewer 1 Report

The Authors submitted to Molecules a paper about an extract from seeds of Annona cherimola. They characterized this extract and tested its biological effects in tumoral cell line by two assays (Apoptosis assay and Anchored growth independent Assay). They affirmed that this extract inhibits the capacity for unlimited proliferation of gastric cancer cells, but this is not observed for normal gastric cells GES-1 used as control.
The study could be interesting, the research of natural products as anticancer agent whit limited side effect in healthy cells is necessary, useful, actual and if these products could derive from agrifood waste then there would also be the advantage of valorising the waste.
If the part relating to the preparation and characterization of the extract is exhaustive and well detailed, the same cannot be said for tests on cells.
As regards the apoptosis test, it is not reported in materials and methods as the assessment of Annexin/nucleus in the graph in Fig 1.b was carried out. Furthermore, the images in Fig 1A seem to indicate an incorrect preparation of the control sample. Too few are the control cells that you see in the image. Regarding the Anchored growth independent Assay, the results could be interesting, but the number of colonies compared is extremely small even for a test like this one.
The results of the biological tests are too preliminary and need to be reviewed/explored to make such strong statements as those made by the Authors and further studies, with even adequate standards of the molecules present in the extract, are necessary before thinking about proceeding with in vivo studies.

Author Response

Reviewer 1: Comments/ Answer

We thank the Reviewer for their interest in our work and for helpful comments that will greatly improve the manuscript and we have tried to do our best to respond to the points raised.

As indicated below, we have checked all the comments provided by the Reviewer and have made necessary changes accordingly to their indications:

Comment 1

If the part relating to the preparation and characterization of the extract is exhaustive and well detailed, the same cannot be said for tests on cells.

  1. As regards the apoptosis test, it is not reported in materials and methods as the assessment of Annexin/nucleus in the graph in Fig 1.b was carried out.
  2. Furthermore, the images in Fig 1A seem to indicate an incorrect preparation of the control sample. Too few are the control cells that you see in the image.

Answer 1

  1. Lines 196- 200. The information was added in the manuscript (section Materials and Methods): “Finally, we quantified the apoptotic cells expressed as a ratio (numbers of positive Annexin V cells/number of nucleus cells), the number of cells was determined by the software ZEN-2.1 modulo image analysis. The number of positive annexin V cells was identified in channel 488 nm (over the threshold 11500), and the number of nuclei in channel 405 nm.”
  2. The ratio between the number of cells positive for Annexin V and the total number of nuclei allows us to normalize the quantification to make it comparable between controls and treatments, thus eliminating the effect of calculating only Annexin V. We refer to the working area as ROI (Region of Interest); it is the same analysis region for each of the preparations.

Comment 2

Regarding the Anchored growth independent Assay, the results could be interesting, but the number of colonies compared is extremely small even for a test like this one.

Answer 2

The number of cells seeded in the semi-solid medium (agarose and medium) shows growth above 100 nm after 10 days. These results make it easier to clearly distinguish one cell accumulation from another. However, with higher numbers of cells, the clones start to connect, making it more difficult to count them accurately.

Comment 3

The results of the biological tests are too preliminary and need to be reviewed/explored to make such strong statements as those made by the Authors and further studies, with even adequate standards of the molecules present in the extract, are necessary before thinking about proceeding with in vivo studies.

Answer 3

Thanks for the suggestion, the manuscript was improved as requested in the current review (introduction section was restructured and conclusion was rewritten).

Reviewer 2 Report

The Manuscript entitled “Annona cherimola seed extracts trigger an early apoptosis response
and selective anticlonogenic activity against human gastric carcinoma cell line SNU-1” submitted by the author to be published in Molecules is very nicely written. I have few points which should be addressed before final acceptance:

1.     Annona cherimola being scientific name should be italicized in the title.

2.     Line 56, A. cherimola should be given as full name as appeared first time in manuscript.

3.      Line no. 86, no should be replaced with not

4.     Line no. 170, CaCl2 should be replace with CaCl2

5.     Line 235, A. cherimola should be italicized.

After incorporating all the above suggestions, I believe manuscript is suitable for publication.

Author Response

Reviewer 2: Comments/ Answer

We thank the Reviewer for their interest in our work and for helpful comments that will greatly improve the manuscript and we have tried to do our best to respond to the points raised.

As indicated below, we have checked all the comments provided by the Reviewer and have made necessary changes accordingly to their indications.

Comment 1

Annona cherimola being scientific name should be italicized in the title.

Answer 1

The title- Thanks for the correction, the scientific name was modified.

Comment 2

Line 56, A. cherimola should be given as full name as appeared first time in manuscript.

Answer 2

Line 56- The abbreviated form was changed to the full name.

Comment 3

Line no. 86, no should be replaced with not.

Answer 3

Line 86- Thanks for the correction, the word was changed.

Comment 4

Line no. 170, CaCl2 should be replace with CaCl2

Answer 4

Line 170- The information was corrected in the Materials and Methods section.

Comment 5

Line 235, A. cherimola should be italicized.

Answer 5

Line 235- The scientific name was changed to italics.

Round 2

Reviewer 1 Report

The Authors addressed my comments only partially. As for the analysis of samples for the annexin test, I appreciate the improvements, but in such an assay the control sample cannot have fewer cells than samples treated with apoptosis-inducing molecules. this seems to indicate that something is wrong. It was already clear that the analysis was related to ROI. I continue to have doubts about the anchored growth independent assay as well. The Authors' reply did not add anything to what was already reported in the previous version of the manuscript. Instead I appreciate the changes made in the conclusions regarding what was suggested in the previous review.

Despite my doubts, especially given the validity of the part relating to the preparation and characterization of the extract, if the Editor will consider the paper worthy of publication, I will agree. Advising greater caution for future bioassays

MInor editing of English language is necessary

Author Response

Reviewer 1: Comments/ Answer

Dear reviewer, thank you for your review of our work. We have addressed your comments and have attached our responses for your consideration:

Comment 1

As for the analysis of samples for the annexin test, I appreciate the improvements, but in such an assay the control sample cannot have fewer cells than samples treated with apoptosis-inducing molecules.

Answer 1

The image of control was replaced with a more representative one (Figure 1A, e-h).

Comment 2

I continue to have doubts about the anchored growth independent assay as well. The Authors' reply did not add anything to what was already reported in the previous version of the manuscript.

Answer 2

Line 194. "Figure 1S" was added to the manuscript. Additionally, the determination of the appropriate number of SNU-1 cells for clonogenic assay was added in Supplementary material (Figure 1S). Through this method, we observe the growth dynamics of the cells and pinpoint environments conducive to easy quantification. This approach enables us to readily detect fluctuations in the numbers of clones, facilitating the identification of both increments and decrements. By evaluating diverse cell densities in each well, we gain insights into optimal conditions for observing notable changes in clone populations.